# Arabidopsis CNL receptor SUT1 confers immunity in hydathodes against the vascular pathogen *Xanthomonas campestris* pv. *campestris*

Nanne W. Taks[1], Marieke van Hulten[1], Jeroen A. van Splunter-Berg[1], Sayantani Chatterjee[1], Floris D. Stevens[1], Misha Paauw[1], Sebastian Pfeilmeier[1], Harrold A. van den Burg [1,2]*

1 Molecular Plant Pathology, Swammerdam Institute for Life Sciences, Faculty of Science, University of Amsterdam, Amsterdam, the Netherlands, 2 Rijk Zwaan Zaadteelt en Zaadhandel B.V. De Lier, The Netherlands

* H.A.vandenBurg@uva.nl

## Abstract

Bacterial plant pathogens exploit natural openings, such as pores or wounds, to enter the plant interior and cause disease. Plants guard these openings through defense mechanisms. However, bacteria from the genus *Xanthomonas* have specialized in that they enter their host via a special entry point, the hydathode—an organ at the leaf margin involved in xylem sap guttation. Hydathodes can mount an immune response against bacteria, including non-adapted and adapted pathogens like *X. campestris* pv. *campestris* (Xcc) that cause vascular disease. Previously, it was shown that the RKS1/ZAR1 immune complex confers vascular resistance against Xcc by recognizing XopAC activity, a type III effector (T3E). However, in absence of XopAC recognition, Arabidopsis Col-0 hydathodes still display resistance against Xcc. Here we mapped the causal gene using an inoculation method that promotes Xcc hydathode entry. Using a population of Recombinant Inbred Lines (RILs) of a cross between a susceptible (Oy-0) and resistant accession (Col-0), a major QTL for Xcc resistance was found on the right arm of Chromosome 5 in Col-0. Combining this result with a genome-wide association analysis yielded a single candidate gene encoding a coiled-coil nucleotide-binding leucine-rich repeat (CNL-type) immune receptor protein called SUPPRESSOR OF TOPP4 1 (SUT1). Expression of *SUT1* was confirmed in hydathodes. We reveal that RKS1/ZAR1 and SUT1 confer different levels of Xcc resistance in different tissue types. Both RKS1/ZAR1 and SUT1 are alone sufficient for Xcc resistance in Col-0 hydathodes. However, RKS1/ZAR1 resistance is also effective in tissue types that represent late infection stages, i.e., xylem and mesophyll. In contrast, SUT1 resistance is not effective in the xylem, while weakly additive to RKS1/ZAR1 in the mesophyll. We thus identify a novel *R* gene, *SUT1*, that confers Xcc resistance primarily early in the infection during hydathode colonization.

**Data availability statement:** All relevant data are within the manuscript and its Supporting Information files.

**Funding:** This research was supported by the Topsector T&U TKI programs "Stop natural entry" and "Finding the Achilles' heel of Brassica for Black Rot disease" in partnership with breeding companies Bejo Zaden B.V. and Rijk Zwaan Zaadteelt en Zaadhandel B.V. (grants EZ-2012-02 and TU18024 to H.A.v.d.B.) and by an NWO-XS grant (OCENW.XS24.1.155 to S.P.). The funders had no role in study design, data collection and analysis, decision to publish, or preparation of the manuscript.

**Competing interests:** I have read the journal's policy and the authors of this manuscript have the following competing interests: H.A.v.d.B. is currently an employee of Rijk Zwaan. This work was supported in part by a grant from Rijk Zwaan and Bejo Zaden. Both companies had no role in the design of the study, data collection, and writing of the manuscript.

## Author summary

Black rot disease, caused by the bacterial pathogen *Xanthomonas campestris* pv. *campestris* (Xcc), is an economically important disease of cabbage crops. Xcc is unique in that it enters the plant interior through specialized organs at the leaf margin. These structures called hydathodes contain water pores and are involved in root pressure regulation. Although we know that hydathodes can mount an immune response against these bacteria, specific immune receptors still need to be discovered. Here we use the model plant *Arabidopsis thaliana* to map an hydathode-effective resistance gene. By screening two different Arabidopsis populations, we could map a single gene, *SUT1,* that is involved in this resistance. SUT1 restricts the early hydathode colonization by Xcc thus suppressing disease progression. Interestingly, SUT1 resistance was not effective in the plant vascular system and leaf mesophyll tissue, which the bacteria subsequently colonize. Therefore, this study provides a new insight into the role of hydathodes in anti-bacterial resistance in plants and opens the door for research on tissue- and organ-specific resistance mechanisms.

## Introduction

Plant pathogenic microbes employ different strategies to invade their host and induce disease. Foliar bacterial pathogens primarily exploit natural openings of plants, such as pores involved in gas exchange (stomata) or in water and nutrient homeostasis (hydathodes). Consequently, plants have evolved a plethora of immune mechanisms to guard these openings and to respond to invading pathogens [1,2]. Among these entry points, stomata are arguably the best studied in the context of plant immunity. Several immune mechanisms have been elucidated, including the sensing of microbe-associated molecular patterns (MAMPs) [3]. Upon MAMP perception, plants activate a signaling cascade that includes pre-invasive closure of the stomatal pores followed by post-invasive opening [4–6]. While stomata provide access to the leaf apoplast, hydathodes represent an entry point for highly specialized bacterial pathogens that colonize the leaf vasculature [7–9]. The hydathode anatomy is composed of an enclosed extracellular cavity interspersed with small mesophyll-like epithem cells. The cavity is connected to the exterior via stomata-like openings in the epidermis called water pores [10,11]. Water pores allow xylem sap exudation, a process known as guttation, in conditions when the root pressure exceeds water evaporation by transpiration [12]. The cavity is connected to endings of the xylem trachea. The epithem tissue appears to act as an active filter of the guttation fluid, reabsorbing salts and nutrients prior to extrusion [12]. Interestingly, the water pores do not completely close in response to pathogens [7,13], providing nearly unrestricted access to the hydathode cavity. Several pathogenic *Xanthomonas* spp. are known to invade their hosts via hydathodes [7,8,14]. Despite the absence of pre-invasive immunity, a post-invasive immune response is mounted in hydathodes against hydathode-adapted and non-adapted bacterial pathogens [15],

which restricts bacterial proliferation in the hydathode cavity and thereby reduces the number of vascular outbreaks. Intriguingly, although general immune signaling hubs have been confirmed to be involved in this hydathode immune response, knowledge on immune receptors that act in hydathodes remains limited.

Plant immune receptors can be classified as cell-surface and intracellular receptors [16,17]. While cell-surface pattern recognition receptors (PRRs) generally detect well-conserved microbe-associated molecular patterns (MAMPs), intracellular nucleotide-binding leucine-rich repeat (NLR) receptors respond directly or indirectly to the mode of action of pathogen-derived effector proteins. These effectors promote virulence upon delivery into the host cell to manipulate cellular processes including plant immunity [18]. Plant NLRs can be subdivided into three groups based on their N-terminal domain, that is, Coiled-coil (CC) NLRs (CNLs), Toll/Interleukin-1 receptor/Resistance (TIR) NLRs (TNLs), and RPW8-like CC domain (RPW8) helper NLRs (RNLs) [19]. While some NLRs activate immune responses by themselves through oligomerization into a so-called resistosome complex [20], others rely on helper NLRs or are part of an NLR network that triggers immune signaling [17]. The evolution of plant immune receptor networks and pathogen immune suppressors has shaped the landscape of most plant-pathogen interactions [16].

Both plant immune receptors and pathogen effectors have been studied extensively in *Arabidopsis thaliana* (hereafter Arabidopsis), in particular using the stomata-invading bacterium *Pseudomonas syringae* [21]. For hydathode-infecting bacteria, such as *Xanthomonas campestris* pv. *campestris* (Xcc) [9], only a few resistance mechanisms have been studied. For example, the RESISTANCE RELATED KINASE 1 (RKS1)/HOPZ-ACTIVATED RESISTANCE (ZAR1) immune complex recognizes uridylation of the pseudokinase PBL2 by the Xcc effector XopAC (AvrAC) [22,23]. This immune complex confers resistance to Xcc already in hydathodes [7,15] as well as later during the infection when the bacteria spread systemically along the xylem vessels [23] and eventually breakout towards the mesophyll [24]. However, many Arabidopsis accessions, including Col-0, still display a considerable level of resistance in hydathodes against Xcc strains that lack *xopAC* [15], indicating that additional pathogen perception mechanism are present in these accessions.

This notion is corroborated by the fact that plant lines lacking the PRR co-receptors BRI1-ASSOCIATED RECEPTOR KINASE 1 (BAK1) and its close homolog BKK1 (BAK1-LIKE 1, SERK4) are more susceptible to hydathode colonization by bacterial pathogens [15]. In addition, the immune signaling hub formed by the proteins ENHANCED DISEASE SUSCEPTIBILITY 1 (EDS1) – PHYTOALEXIN DEFICIENT 4 (PAD4) – ACTIVATED DISEASE RESISTANCE 1 (ADR1) was shown to be needed for restricting hydathode colonization by Xcc [15]. Hence, it becomes apparent that another effective, post-invasive immune response exists in Arabidopsis Col-0 against adapted bacteria in hydathodes, yet the causal receptor for this immune recognition remains elusive. To fill this knowledge gap on hydathode immunity, we set out to identify the causal resistance gene.

## Results

### A QTL on Chromosome 5 is associated with resistance against Xcc ΔxopAC

Earlier, we showed that the Arabidopsis accession Col-0 shows a high level of resistance to the vascular bacterial pathogen Xcc8004 in guttation-based infections and that this response was independent of XopAC recognition by RKS1/ZAR1 [15]. Furthermore, the same study showed that the accession Oy-0 is susceptible to the same mutant bacterial strain (Xcc8004 ΔxopAC), meaning that we can use a segregating population between Oy-0×Col-0 to map the causal gene(s) for this resistance. To this end, we made use of an existing Arabidopsis population of Recombinant Inbred Lines (RILs) between Oy-0×Col-0, composed of 470 RILs [25]. In total, we screened for resistance in 165 RILs of this population that showed the most complete annotation of markers to the parental genomes, including the two parental lines Oy-0 and Col-0, using a bioluminescent reporter strain (Tn5:*lux*) of Xcc8004 ΔxopAC [15,26]. For each plant, the three most symptomatic leaves were sampled 14 days post inoculation (dpi) and bacterial colonization was visualized using light-sensitive films (Fig 1A). The level of resistance was then scored using a luminescence index ordinal scale [26] and normalized to the Oy-0 control for each experiment (Fig 1B and 1C). Using an Expectation-maximization (EM) model for a single QTL

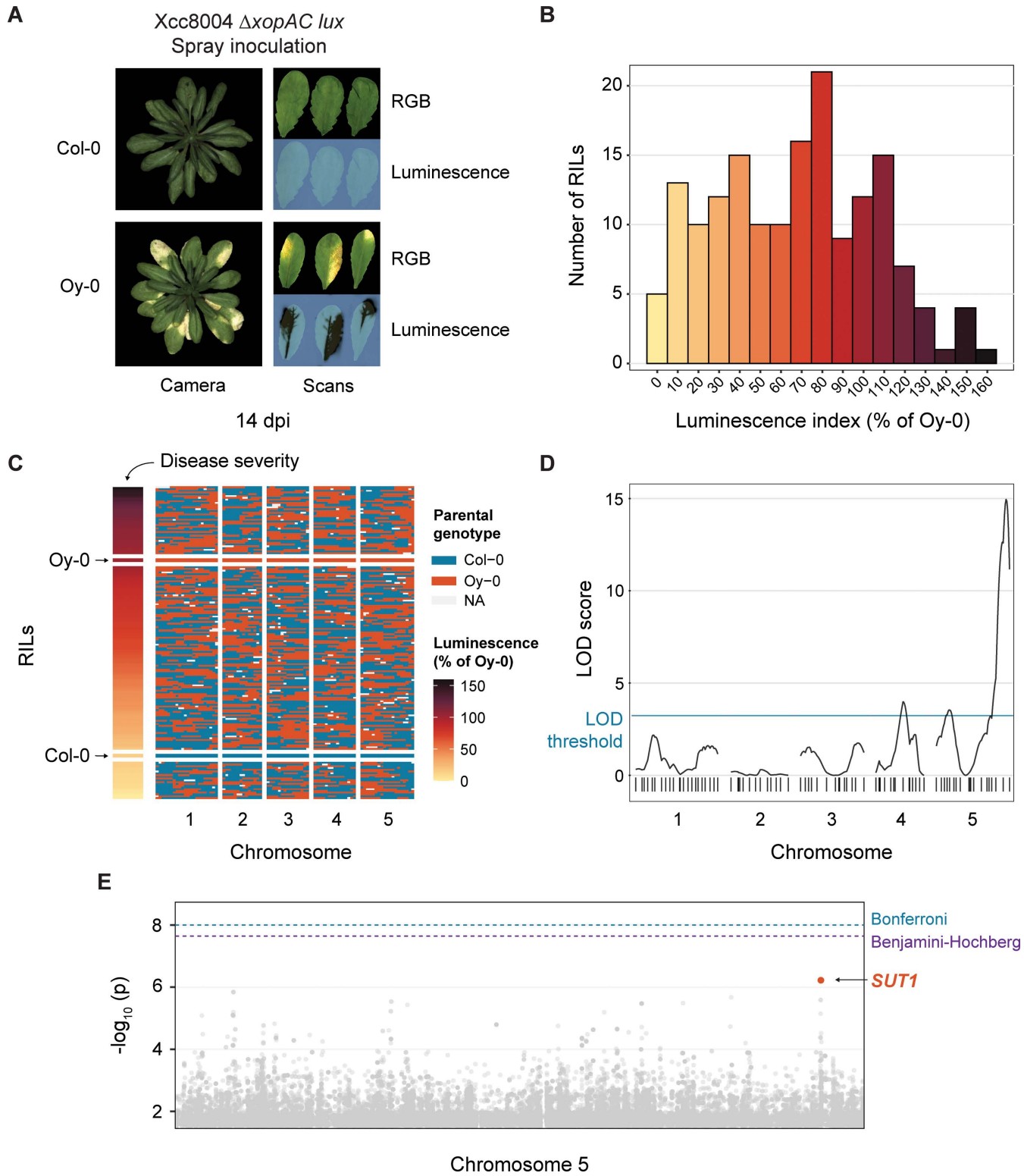

**Fig 1. Mapping of *SUT1* as a candidate *R* gene against Xcc8004 Δ*xopAC* in Arabidopsis Col-0. A)** Disease symptoms and bacterial luminescence in the resistant Col-0 and susceptible Oy-0 accessions following spray inoculation with bioluminescent Xcc8004 Δ*xopAC*. Leaves shown are representative of spray inoculation with either Tn*5:lux* or Tn*7:lux* tagged reporter strains. Bacterial spread was visualized using light-sensitive film 14 days post

inoculation (dpi). **B)** Histogram depicting the variation in luminescence index score 14 dpi (normalized to Oy-0) for all 165 Oy-0×Col-0 Recombinant Inbred Lines (RILs) following spray inoculation with Xcc8004 Δ*xopAC* Tn*5:lux*. **C)** Population structure of 165 RILs in the Oy-0×Col-0 RIL population, ordered by luminescence index score at 14 dpi (normalized to Oy-0) shown in (B). **D)** QTL graph showing LOD scores for each of the 85 markers in the RIL population. LOD significance threshold determined by permutation (n=1000) at 2.49 **E)** Manhattan plot of Chromosome 5 from a screen of 321 accessions present in the HapMap collection. Highlighted SNPs in orange are located within the coding sequence of *SUT1*. False Discovery Rate (FDR) thresholds set at -log$_{10}$ (p) = 8 (Bonferroni) and -log$_{10}$ (p) = 7.29 (Benjamini-Hochberg). See also S1 Fig for the full Manhattan plot.

[27] and LOD threshold set at 2.49 by permutation (n=1000), three QTLs were identified, that is, one at Chromosome 4 and two on Chromosome 5 (Fig 1D). These QTLs showed an explained variance (EV) of 10, 11 and 32%, respectively. The major QTL at the end of the right arm of Chromosome 5 had an LOD score of 14.92 and thus displayed a strong genetic association with the resistance phenotype in the Oy-0×Col-0 RIL population.

## Genome-wide association study (GWAS) identified candidate *R* gene *SUT1*

To identify candidate genes in the major QTL on Chromosome 5, a GWAS was performed with the same Xcc8004 Δ*xopAC* Tn*5:lux* reporter strain on the Arabidopsis HapMap population [28]. In total 321 accessions from this populations were evaluated for Xcc disease susceptibility 14 dpi and the data obtained was normalized to the Oy-0 control to control for batch effects. GWA analysis was performed on phenotypic data using an online GWA portal, based on an imputed SNP dataset of over five million SNPs [29]. Unexpectedly, no SNPs were identified with a significant correlation to the disease phenotype on Chromosome 5 (Figs 1E and S1). However, several SNPs that were close to the False Discovery Rate (FDR)-corrected significance threshold coincided with the major QTL (Fig 1E), suggesting an association with the Col-0 disease resistance. These SNPs were located in the coding sequence of the gene At5g63020, which codes for SUPPRESSOR OF TOPP4 (SUT1), a previously characterized CNL involved in plant immunity [30].

## SUT1 is involved in hydathode resistance against Xcc

To further investigate the role of SUT1 in plant immunity in Col-0 and Oy-0, we aligned the respective *SUT1* alleles using recently published whole genome assemblies of the two accessions [31] and confirmed them by Sanger sequencing of the Col-0 and Oy-0 accessions used in our laboratory. The sequence alignment identified SNPs in the *SUT1*$^{Oy-0}$ coding sequence including six non-synonymous changes that result in amino acid substitution (Figs 2A and S2). In addition, one single base insertion leads to a frameshift and a predicted three-residue peptide extension of SUT1$^{Oy-0}$. All but one of the predicted amino acid changes are in the leucine-rich repeat (LRR) domain, which is known to act as a protein-protein interaction surface [32] for target sensing and ligand binding [20,33]. These sequence differences between *SUT1*$^{Col-0}$ and *SUT1*$^{Oy-0}$ could thus explain the difference in Xcc disease resistance between Oy-0 and Col-0

To assess if a *SUT1* null allele in Col-0 would lead to loss-of-resistance, two known *sut1* T-DNA insertion mutants, *sut1−8* and *sut1−9,* were tested for increased Xcc disease susceptibility (Fig 2B). Previously, these two T-DNA insertion lines were shown to result in recovery of the *topp4−1* dominant autoimmune phenotype [30], implying that SUT1 is no longer functional in both mutants. To confirm a role of SUT1 in Xcc resistance, both mutants were spray inoculated with Xcc8004 Δ*xopAC* Tn*7:lux* and the level of resistance was assessed both early in the infection in hydathodes (7 dpi) and at late stages during leaf colonization (14 dpi). Compared to the wildtype Col-0 control, both *sut1−8* and *sut1−9* showed an increased number of severe hydathode infections at 7 dpi (Fig 2C and 2D), and enhanced leaf chlorosis (Fig 2C and 2E) combined with increased bacterial spread at 14 dpi (Fig 2C and 2F). During the early stage of hydathode colonization, the *sut1* mutants showed an intermediate level of susceptibility in comparison to the susceptible controls (i) *bak1-5;bkk1-1* (*bb*) in the Col-0 background and (ii) the susceptible accession Oy-0. Furthermore, disease symptom development was more severe in the Oy-0 plants than in the *sut1* knockout lines, which is consistent with the fact that two minor QTLs with a significant LOD score were found in the RIL population as well (Fig 1D). Possibly, these two loci are as well involved

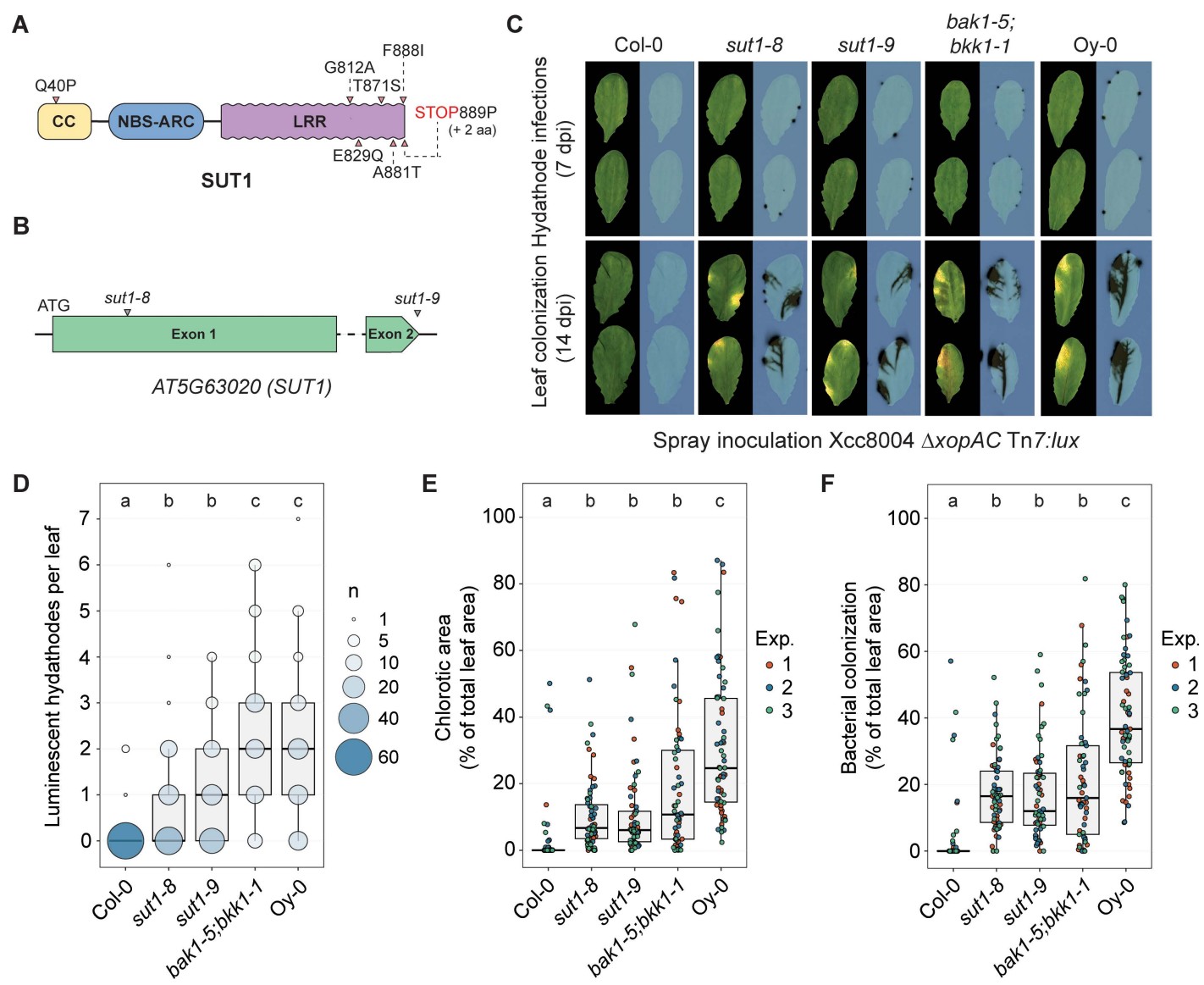

**Fig 2. SUT1 contributes to hydathode resistance against Xcc8004 ΔxopAC. A)** Predicted SUT1 protein model. Arrows depict predicted amino acid differences between Oy-0 vs. Col-0. **B)** *SUT1* gene model depicting the Col-0 allele and approximate integration site of the *sut1-8* and *sut1-9* T-DNA insertions. **C)** Spray inoculation of *sut1-8* and *sut1-9* with Xcc8004 ΔxopAC Tn7:lux. Wildtype background (Col-0), *bak1-5;bkk1-1* (Col-0) and the accession Oy-0 were included as controls. *Top,* visualization of high density hydathode colonization using light sensitive films (7 dpi); *bottom*, visualization of the systemic spread of Xcc and disease symptom development (14 dpi). **D,E,F)** Number of infected hydathodes 7 dpi (**D**), chlorotic area 14 dpi (**E**) and bacterial colonization 14 dpi (**F**) per leaf for each line shown in panel C following spray inoculation with Xcc8004 ΔxopAC Tn7:lux (n=21 leaves per experiment). In panel D, the circle size depicts the number (#) of leaves with a certain score (y-axis) from three independent repetitions. Significance letters from non-parametric Kruskal-Wallis test with Dunn's Post-Hoc test, p-value threshold=0.05.

in disease resistance besides *SUT1* in Col-0. A previous study showed that SUT1 is needed for dominant autoimmune phenotype in the Col-0 mutant background caused by a specific protein variant of the TYPE 1 PROTEIN PHOPSHATASE 4 (TOPP4), suggesting that the NLR SUT1 guards the TOPP4 protein activity [30]. To infer if TOPP4 itself is also important for Xcc resistance in Col-0, a previously characterized *topp4* knockout line was inoculated with Xcc8004 ΔxopAC Tn7:lux. A minor but significant increase in the number of infected hydathodes was seen in the *topp4* knockout line, but no

increase was seen in bacterial spread across the leaf or leaf chlorosis during later infection stages (S3 Fig). Apparently, SUT1 immunity to Xcc does not depend one-on-one on guarding TOPP4. Future work should reveal whether other TOPP family members have a role in SUT1 resistance against Xcc. We thus conclude that SUT1 restricts Xcc proliferation inside hydathodes, thereby reducing both Black rot disease symptom development and the systemic spread of Xcc along the leaf vasculature.

### *SUT1* is expressed in different cell types including the epithem

As SUT1 confers resistance already in the hydathodes, we hypothesized that the *SUT1* gene is expressed in the hydathode epithem cells. To further investigate this idea, transgenic reporter lines were generated that express the *β-glucuronidase (GUS)* reporter gene from the *SUT1* promoter using the 2.1 kb DNA fragment upstream of the *SUT1* start codon for histo-chemical staining of whole leaves. GUS straining was detected in the entire leaf of 4-week-old T1 transgenic plants with a more intense blue staining of the hydathodes (Figs 3A and S4). As others reported earlier that GUS staining of hydath-odes can be the result of false positive staining [34], an independent reporter system was generated as well that gives YFP fluorescence in nuclei, *pSUT1::NLS-3×mVENUS* [35]. As a reference for epithem cells, we used our epithem marker line *pPUP1::RCI2A-tdTomato* [15], which displays red fluorescence at the epithem cell membrane (Fig 3B). Col-0 plants expressing *pSUT1::NLS-3×mVENUS* showed a fluorescent YFP signal both in the nuclei of epithem cells and throughout the mesophyll tissue (Figs 3C and S5). These findings are supported by several studies on single-cell transcriptomics that indicate expression of *SUT1* in different cell types [36], including the mesophyll and epithem [12,37]. Taken together, these data confirm *SUT1* expression in the epithem cells within hydathodes.

### SUT1-mediated resistance is ineffective when hydathodes are bypassed

As SUT1 already provides resistance to Xcc already during the first stage of infection, we wondered whether SUT1 also contributes to resistance (a) at later stages of the infection or (b) upon wound inoculation when the bacteria invade the leaf vasculature. To this end, clipping inoculations were performed to provide the bacteria direct access to the midvein and lateral veins near the leaf apex, thereby bypassing the hydathodes. We quantified both the bacterial spread along the midvein and the relative chlorotic leaf area (as a proxy of disease symptom development) (Fig 4A). Strikingly, SUT1 resis-tance was lost when the hydathodes were bypassed. First, we found no difference in the leaf chlorosis between the *sut1* mutants and the wildtype control (Fig 4B). Second, the total distance that the bacteria can progress along the midvein was apparently also unaffected in *sut1−9* compared to the wildtype control Col-0 (Fig 4C). This tissue-specific response differs to RKS1/ZAR1 immunity that is also effective in the vasculature, as shown by clipping inoculation of Xcc8004 WT Tn*5:lux* in Col-0 and both *rks1−1* and *zar1−1* knockout lines (S6 Fig); others reported earlier a similar finding when Xcc was intro-duced into the midvein by pin inoculations [23,38]. Overall, these results indicate that RKS1/ZAR1 and SUT1 display a different resistance level against an Xcc infection in the different tissue types.

### *RKS1/ZAR1* and *SUT1* confer effective resistance in different plant tissues

As we detected *SUT1* expression throughout the leaf, but no effective *SUT1* resistance outside hydathodes, we further examined if *RKS1/ZAR1* and *SUT1* confer additive resistance in the different plant tissues. To this end, *rks1/sut1* double knockout mutants were generated in Arabidopsis Col-0 using the CRISPR/zCas9i toolkit [39] (S7 Fig). Next, we evalu-ate the resistance level that *RKS1/ZAR1* and *SUT1* provide at different stages of the infection using the wildtype strain Xcc8004 Tn*7:lux* strain. To this end, three different inoculation methods were used, that is, spray inoculations (entry via hydathodes), clip inoculations (to access xylem vessels) and syringe infiltrations (to provide direct access to the meso-phyll). Resistance against Xcc was examined in the wildtype Arabidopsis Col-0 (with functional *RKS1/ZAR1* and *SUT1*), plants defective for one of two *R* genes (i.e., *rks1−1* and *sut1−9*) and plants defective in both (i.e., *rks1/sut1* and Oy-0). As Oy-0 is known to be susceptible to Xcc8004 secreting T3E XopAC [15], we assumed that this accession lacks both

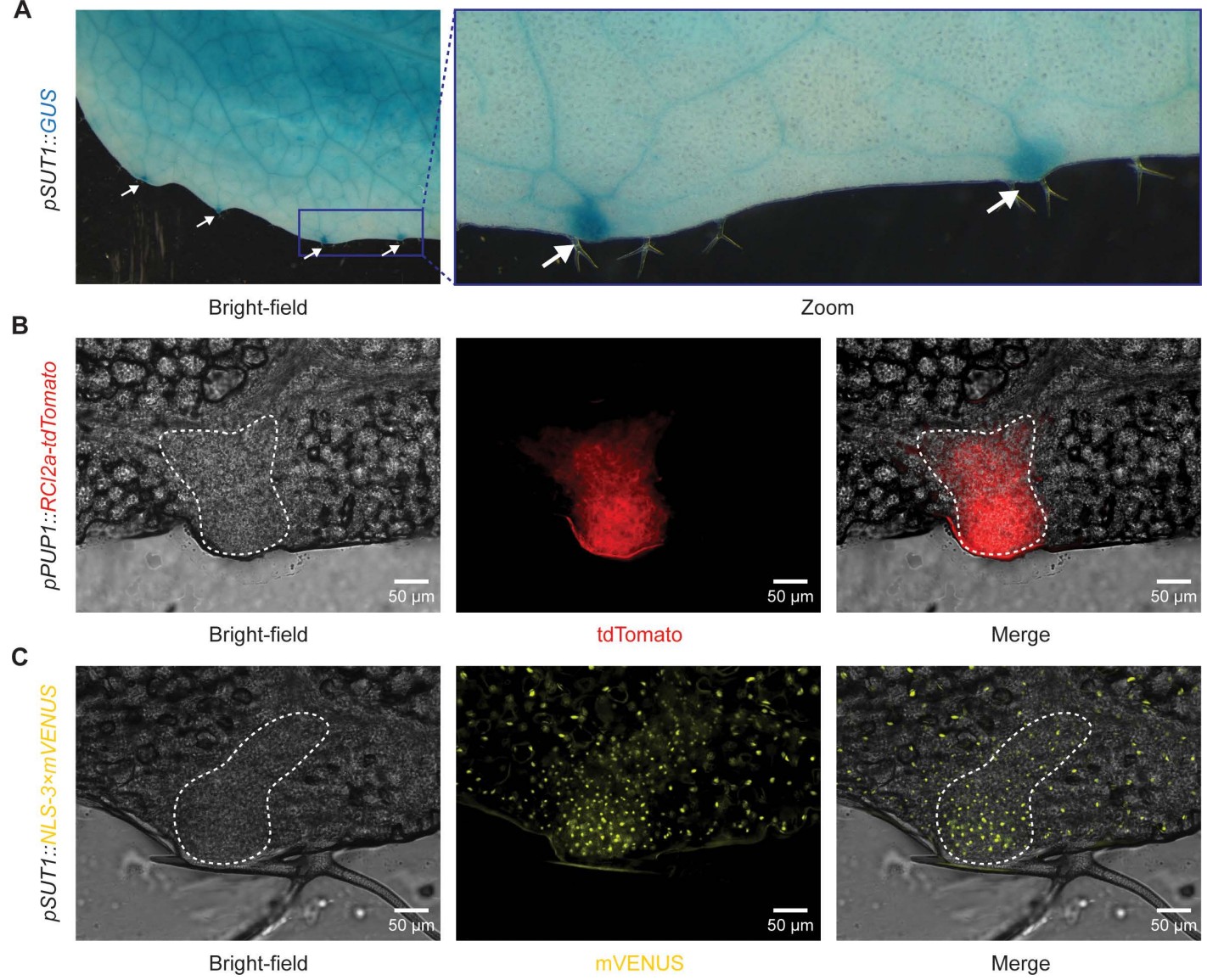

**Fig 3. SUT1 is expressed in epithem cells within hydathodes. A)** Arabidopsis *pSUT1::GUS* lines stained with X-Gluc show *SUT1* expression over the entire leaf (4-week-old plants). GUS staining is stronger in the hydathodes along the leaf margin (white arrows). **B)** Arabidopsis epithem marker line *pPUP1::RCI2a-tdTomato* shows tdTomato localization in epithem cell membrane within hydathodes. Epithem tissue can be distinguished from the mesophyll cells in the bright-field image (white dashed line). **C)** Arabidopsis *pSUT1::NLS3×mVENUS* T2 line shows mVENUS accumulation in nuclei of mesophyll and epithem cells. Epithem cells can be distinguished from mesophyll cells in the bright-field image as a region with dense small cells on the leaf margin (white dashed line). Images were taken of epidermal peels and are representative of three independent lines (S5 Fig).

a functional *RKS1/ZAR1* and *SUT1*. In spray inoculation experiments that mimic natural infection, we found the *rks1–1* and *sut1–9* single mutants displayed a similar level of resistance against Xcc, both in hydathodes (Fig 5A and 5B, 7 dpi) and when the bacteria were spreading along the leaf vasculature (Fig 5A and 5C, 14 dpi). Therefore, *RKS1/ZAR1* and *SUT1* independently confer resistance in the hydathodes thereby reducing Xcc outbreak towards the xylem. The *rks1/sut1* double mutants showed a similar level of susceptibility as the Oy-0 control, indicating that *RKS1/ZAR1* and *SUT1* together represent the main resistance genes in Col-0 that halt Xcc in the initial phase of the infection.

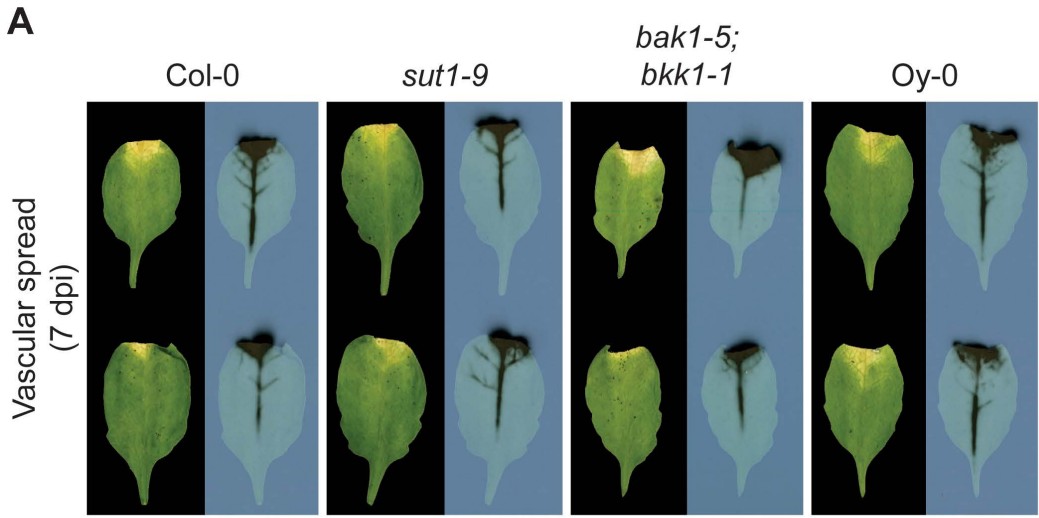

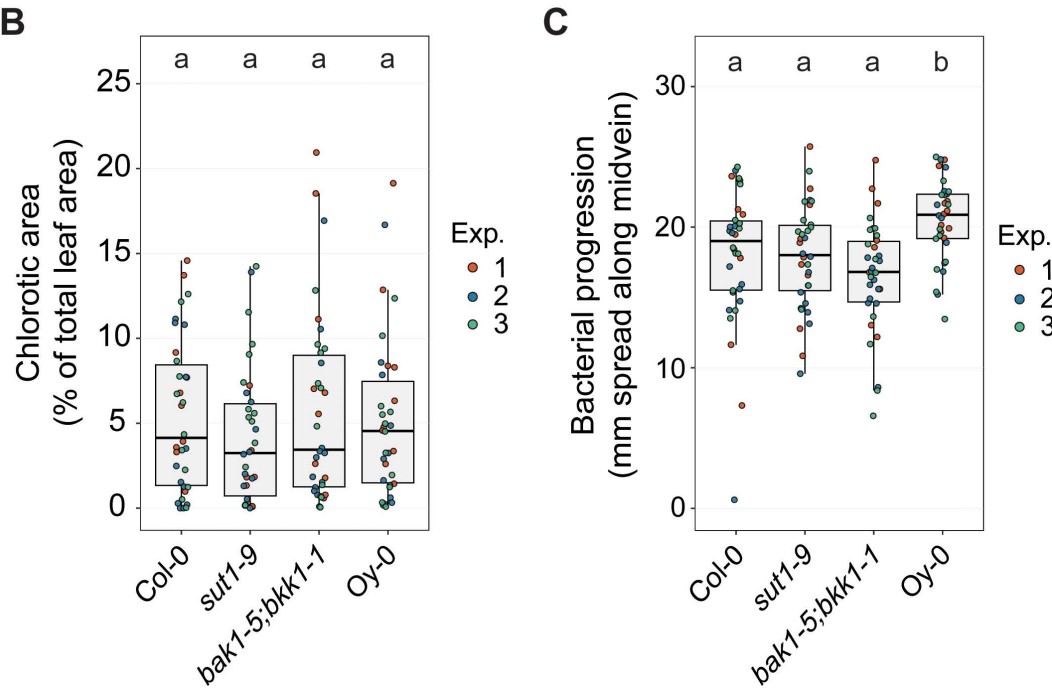

**Fig 4. SUT1 resistance is compromised when the Xcc inoculation bypasses hydathode colonization. A)** Disease symptoms and bacterial spread in Col-0, *sut1-9*, *bak1-1;bkk1-5* and Oy-0 leaves at 7 dpi following clip inoculation with Xcc8004 Δ*xopAC* Tn7:*lux*. Bacterial progression is quantified as the spread of the luminescence signal along the midvein. **B,C)** Chlorotic leaf area (**B**) and bacterial spread along the midvein (**C**) 7 dpi following clip inoculation with Xcc8004 Δ*xopAC* Tn7:*lux*. SUT1 resistance against Xcc is apparently ineffective in the vasculature of Col-0 when the hydathodes are bypassed and the bacteria are introduced using leaf clipping. Significance letters from a two-way ANOVA with Tukey Post-Hoc test, p-value threshold = 0.05 (n = 12 leaves per experiment).

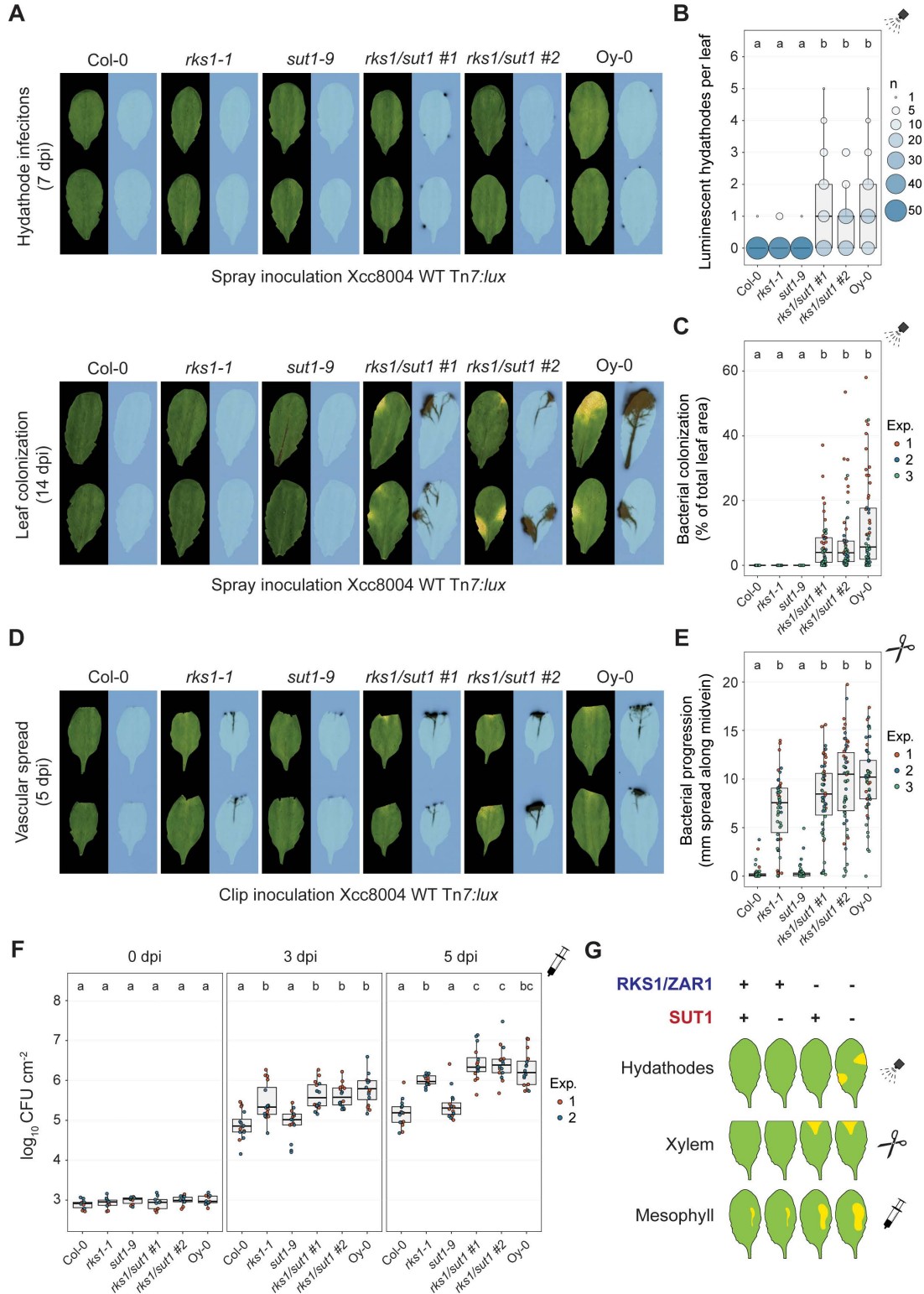

**Fig 5. *RKS1/ZAR1* and *SUT1* confer resistance to Xcc in different stages of infection. A)** Spray inoculation with Xcc8004 Tn7:*lux* on *rks1-1* and *sut1-9* single mutants and two independent *rks1/sut1* CRISPR knockout lines with wildtype background Col-0 and accession Oy-0 as controls. *Top,*

visualization of Xcc hydathode colonization using light sensitive films (7 dpi); *bottom*, visualization of the systemic spread of Xcc and disease symptom development (14 dpi). **B,C)** Quantification of panel A as number of infected hydathodes per leaf at 7 dpi (**B**) and bacterial colonization (**C**) at 14 dpi (n = 18 leaves per experiment). **D)** Clip inoculation with Xcc8004 Tn7:*lux* on the same plant lines used in panel A. **E)** Quantification of bacterial progression along the midvein of panel D at 5 dpi (n = 18 leaves per experiment). **F)** Colony forming units at 0, 3 and 5 days post syringe inoculation of Xcc8004 WT Tn7:*lux* in the apoplast of the plant lines used in panel A (n = 8 samples per experiment. **G)** Schematic model indicating the contribution to resistance by *RKS1/ZAR1* and *SUT1* in different leaf tissues.

When the bacteria were directly introduced into the xylem vessels by clipping, it was noted that *RKS1/ZAR1* effectively prevented Xcc vascular spread (Fig 5D and 5E). In contrast, *SUT1* resistance appears not to act in the xylem, as the *rks1–1* plants showed a similar vascular spread as the *rks1/sut1* double mutants and the Oy-0 control (Fig 5D and 5E), confirming our earlier observations (Fig 4). Therefore, *RKS1/ZAR1* is effective in halting an Xcc infection even when the hydathodes are bypassed, whereas *SUT1* is not. To further assess the resistance level at the late infection stage when Xcc spreads into the mesophyll tissue, direct mesophyll inoculations were performed using syringe infiltrations. Plants expressing a functional *RKS1/ZAR1* showed a strong reduction in bacterial proliferation compared to Oy-0 (Fig 5F). Similar to the xylem vessels, we found that the *rks1–1* plants showed significantly more Xcc bacterial proliferation than the *sut1–9* plants, indicating that *RKS1/ZAR1* resistance is also the main source of resistance in the Col-0 mesophyll. In corroboration, at 5 dpi the *rks1/sut1* double mutants displayed a small yet significant increase in Xcc proliferation compared to the *rks1–1* single mutant, suggesting that SUT1 has a minor additive effect on plant immunity in the Col-0 mesophyll against Xcc (Fig 5F). Altogether, these results indicate a differential immune response by *RKS1/ZAR1* and *SUT1* to Xcc infections in different tissue types, as *RKS1/ZAR1* confers resistance in all tested plant tissues, while *SUT1* only provided resistance in hydathodes (Fig 5G).

## Discussion

Here, we characterized a role for Arabidopsis SUT1 as a functional immune receptor in natural infections by Xcc. We showed that SUT1 acts in hydathodes, restricting Xcc colonization in the initial stage of the infection and thereby reducing the subsequent systemic spread of Xcc and with that the onset of disease symptoms. Interestingly, SUT1 resistance was strongly compromised when the hydathodes were bypassed, indicating that SUT1 confers a first layer of defense in the Xcc disease cycle. We found that this role of SUT1 contrasts sharply with the previously characterized RKS1/ZAR1 immune response triggered by XopAC [23,24]. The resistance conferred by this so-called ZAR1-resistosome in different plant tissues can be explained by its role in broad-spectrum resistance against the mesophyll pathogen Pst [40] and the vascular Xcc. So far, no evidence exists for a role of SUT1 in immunity against mesophyll pathogens [30], which might explain why it only confers resistance to Xcc in hydathodes. Yet, our data indicates expression of *SUT1* in hydathodes as well as the mesophyll, which is corroborated by several recent single-cell transcriptomics studies [12,36,37]. Nonetheless, the resistance conferred by SUT1 in the mesophyll against Xcc remains limited compared to the complete resistance found in hydathodes (Fig 5), suggesting that the expression of a putative guardee or bacterial elicitor could be tissue specific. Indications for this can be found in recent work on the HrpG virulence regulon of 17 *Xanthomonas* spp., which revealed that not all virulence factors are controlled by HrpG, highlighting the possibility of temporal regulation of secreted virulence factors by *Xanthomonas* during infection [41]. In addition, the absence of vascular immunity conferred by SUT1 remains atypical, as different vascular immune responses against Xcc have been identified, such as XOPAM ACTIVATED RESISTANCE 1(AMAR1)-mediated hypersensitive response through interaction with the T3E XopAM [42] and MAPK signaling cascades that lead to reduced vascular spread and symptom development when Xcc is directly introduced into the vasculature [22]. While expression of *SUT1* along the leaf vasculature is confirmed by this work (Fig 3A) as well as other studies using single-cell transcriptomics [36,37], xylem sap is conducted via dead cells and *SUT1* is therefore likely expressed in the neighboring xylem parenchyma cells that surround the xylem or in other vascular cell types.. More

general, how *R* gene resistance can be effective against xylem-infecting pathogens, as conferred here by *RKS1/ZAR1,* remains unclear and is a topic of ongoing discussion [36].

As *SUT1* encodes an intracellular immune receptor, our data support the notion that Xcc actively manipulates the epithem tissue to gain access to the plant vasculature. Importantly, the causal agent for SUT1 immune activation remains unknown albeit that an indirect activation mechanism has been proposed where SUT1 guards the TOPP4 phosphatase has been proposed in the context of autoimmunity [30]. While we did not find evidence for this TOPP4 guard-guardee model in the interaction with Xcc (S3 Fig), the two closest homologs of SUT1 in Arabidopsis, RESISTANT TO P. SYRINGAE 5 (RPS5) and SUPPRESSOR OF MKK1 MKK2 2 (SUMM2) [30,43], have both been described as 'guards' in plant-pathogen interactions [44,45], which argues in favor of the possibility of a similar guard function of SUT1. In addition, AlphaFold3 has very recently been leveraged to distinguish between helper and sensor functions of plant NLRs, which could also be used to predict the function of SUT1 in pathogen recognition [46].

Indications for the immune elicitor could be found in the putative guardee TOPP4, which is part of an Arabidopsis gene family of highly similar protein phosphatases called TOPP1 to TOPP9. This family is involved in abscisic acid (ABA) signaling and leaf water homeostasis [47,48]. Interestingly, multiple TOPP family members are targeted by the *Pseudomonas syringae* T3E AvrE during infection to hijack ABA signaling, thereby suppressing the re-opening of the stomata upon bacterial entry to induce water soaking of the leaf apoplast—a phenomenon that supports bacterial colonization of the leaf apoplast [49–51]. Notably, Xcc8004 is known to employ the T3E XopAM, which belongs to the same protein superfamily of conserved bacterial effectors as *Pseudomonas* AvrE [52]. Nonetheless, the AvrE superfamily members can exhibit a wide variety of functions [53] and recent work using a yeast two-hybrid screen on an Arabidopsis cDNA library did not reveal a protein-protein interaction between XopAM and individual TOPPs [42]. Although the elicitor of SUT1-mediated resistance has not been identified, it might be associated with manipulation of TOPPs and therefore ABA signaling.

When it comes to hydathode colonization, we see an intermediate level of resistance in the *sut1* knockout lines in the Col-0 background compared to the susceptible accession Oy-0. This indicates the presence of additional resistance mechanisms in Col-0 against Xcc in hydathodes. Indeed, we observed two additional minor QTLs on Chromosome 4, which were not further fine mapped in this study. The underlying genes for these QTLs might encode additional NLRs, receptor-like proteins (RLPs), receptor-like kinases (RLKs) or guardees of SUT1, some of which are even known to be specifically expressed in hydathodes [54]. It seems that the resistance phenotype seen in Col-0, in absence of RKS1/ZAR1-mediated resistance, can still be based on several and possibly disconnected resistance mechanisms of which SUT1 resistance appears to be a major contributor. The existence of a wide array of immune receptors in Arabidopsis holds promise for application of these genes in *Brassica* crops to ensure durable resistance using a transgene approach. As was shown in other crop-pathogen interactions, *R* gene stacking is less readily overcome by pathogens than the implementation of single *R* genes [55–57].

The identification of *SUT1* was made possible through the RIL analysis, as the GWA analysis itself did not lead to significantly correlating SNPs in *SUT1*. In these analyses, assessing disease severity relied on luminescence index scoring at 14 dpi [26], which could have reduced the likelihood of identifying resistance mechanisms in the early stages of infection in these large natural populations. For this reason, our lab has put efforts into the development of quantitative image analysis methods for bacterial bioluminescence [58]. In addition, we have developed a non-invasive imaging method to track bacterial infections in Arabidopsis rosettes over time, from hydathodes to mesophyll [59]. Work is in progress to repeat the GWA study using the improved temporal resolution offered by this non-invasive imaging setup, thus increasing the likelihood of identifying additional genetic factors in Arabidopsis that play a role in hydathode resistance against Xcc.

In summary, we identified a CNL-type immune receptor, SUT1, that acts in hydathodes where it reduces Xcc virulence by limiting the initial hydathode colonization and thereby the subsequent disease progression. This SUT1 resistance against Xcc seems to be limited to hydathodes, as the efficacy of the *SUT1* gene was lost when the initial hydathode

barrier was breached. Future studies should aim to solve how SUT1 immunity is activated by Xcc, although SUT1 likely acts as a guard of one or more TOPPs or other plant phosphatases.

## Materials and methods

### Plant growth conditions

*Arabidopsis thaliana* lines used are listed in S1 Table. For all experiments, seeds were stratified for three to four days on wet filter paper at 4 °C in the dark, and then sown in 40-pot trays (Desh Plantpak, Tray Danish size 40 cell (8x5)) in slitpots (Pöppelmann Teku, S 5,5 LB), using potting soil (Jongkind Substrates, Hol80 zaaigrond Nr 1) to which 500 ml Entonem suspension (Koppert Biological Systems, approximately 1.6 x 10$^6$ third stage *Steinernema feltiae* nematode larvae) was added per tray to prevent black fly multiplication in growth chambers. Hereafter, trays were placed in standard short day (SD) conditions (11/13h light/dark, 22 °C, RH 70%). For the first five days after sowing, trays were covered with transparent plastic domes to ensure seedling growth.

### Generation of Arabidopsis reporter lines

Primer sequences and plasmids used are listed in S1 Table. The plasmids to create the *pSUT1::GUS* and *pSUT1::NLS-3×mVENUS* reporter lines in Arabidopsis were assembled using the GreenGate system [60]. First, a 2.1 kb fragment upstream of the start codon of *SUT1* (*AT5G63020*) in Col-0, which also covers the 119 bp 5'UTR indicated by RNA-seq data [31], was synthesized with external *Bsa*I sites (Eurofins Genomics). As the entire adjacent gene (AT5G63030) falls within this region, four potential start codons were modified. Next, the *NLS-3×mVENUS* module was amplified from pGreenII-229-NLS-3×mVENUS [35] with added *Bsa*I sites and cloned into pGGA000. Final binary vectors were then assembled in the backbone pGGZ003 with either *GUS* or *NLS-3×mVENUS* as CDS module using a multisite Goldengate reaction, leading to pGGZ_pSUT1_GUS_tUBQ10_BastaR and pGGZ_SUT1_NLS-3×mVENUS_tUBQ10_BastaR. These plasmids were electroporated into *Agrobacterium tumefaciens* strain GV3101 carrying pSOUP and Arabidopsis Col-0 was transformed using the floral dip method [61]. Two independent GUS lines and three independent *NLS-3×mVENUS* lines were used for imaging.

### Generation of Arabidopsis Col-0 *rks1*/*sut1* knockout lines using CRISPR/zCas9i

Primer sequences and plasmids used are listed in S1 Table. The constructs to generate *rks1*/*sut1* double knockouts were assembled using the zCas9i cloning kit from Addgene (#1000000171). First, for each target gene, two individual guide sequences were designed on the Col-0 alleles using the webtool CRISPR P (v2.0). Next, these four individual guides were cloned into their respective pDGE shuttle vectors using BpiI, followed by assembly in the multiplex destination vector pDGE652 using BsaI. The resulting plasmid was then electroporated into *Agrobacterium tumefaciens* GV3101 and Arabidopsis Col-0 was transformed using the floral dip method [61]. T1 seeds were selected through the *pOLE1:OLE1-RFP* seed coat marker using the Rhodamine filter in Bio-Rad ChemiDoc imager. Next, DNA was extracted from T1 plants using the ThermoFisher Kingfisher Apex and MagMAX Plant DNA Isolation Kit according to manufacturer's instructions. Guide target regions were PCR amplified with Phusion DNA Polymerase using flanking primers and sent for Sanger sequencing and subsequent Synthego ICE analysis. Two individual transformants were selected that exhibited confirmed biallelic knockouts (through early stop codons) in both *RKS1* and *SUT1* (S7 Fig) and were propagated into the next generation. The resulting T2 plants were used for Xcc disease assays (Fig 5).

### Generation of bioluminescent Xcc strain

All bacterial strains, plasmids and oligonucleotides used are listed in S1 Table. Wildtype Xcc8004 was tagged with a double bioluminescence/fluorescence cassette using the mini-Tn7 transposon system [62,63]. Recipient strain Xcc8004

was co-incubated on LB plates with one donor *E. coli* strain (DH5α + pRS-Tn7-*pNPTII::lux-pA1::mTq2* [15,63]) and two helper *E. coli* strains; one carrying Tn7 transposon genes *tnsABCDE* (DH5α + pUX-BF13 [64]) and one carrying a plasmid that facilitates conjugation (HB101 + pRK2073 [65]). Next, cultures were collected and transformants were selected on LB plates with antibiotics (20 μg/ml nitrofurantoin and 50 μg/ml kanamycin) and checked for luminescence using the Chemi-Doc MP imager (Bio-Rad).

### *Xanthomonas* disease assays

Spray inoculations with bioluminescent Xcc8004 WT or Δ*xopAC* (Tn*5:lux* or Tn*7:lux*) strain were performed according to previous studies [15,26]. Briefly, Xcc strains were plated on KADO medium [66] with the appropriate antibiotics and grown for 48–72 hours at 28 °C. Bacteria were collected, washed once with 10 mM $MgSO_4$ and diluted to $OD_{600}$ = 0.1. Silwet L-77 was added to a final concentration of 0.0002% and bacteria were sprayed on 4-to-5-week-old Arabidopsis plants (Preval sprayer, SKU# 0221). Inoculated plants were subjected to two guttation cycles over two days in a climate incubator (Snijders Labs, MC1000) to promote hydathode infection. In these cycles, temperature and relative humidity were raised (25°C, RH 95%) to induce guttation, after which a rapid drop in temperature and humidity (21°C, RH 70%) leads to re-uptake of guttation droplets. Subsequently, plants were placed for an additional 12 days at SD growth conditions (11/13h light/dark, 24–22 °C, RH 70%) in the same incubator. Clipping assays were performed by cutting 5 mm of the apex of each leaf (three leaves per plant) with scissors dipped in bacterial inoculum ($OD_{600}$ = 0.1). Plants were subjected to the same two-day guttation cycle and placed for an additional five days at SD growth conditions in an MC1000 climate chamber.

Syringe inoculations were performed by infiltrating bacterial inoculum ($OD_{600}$ = 0.0001) into the apoplast of each leaf (three leaves per plant) with a needleless 1 ml syringe. Plants were covered with a plastic dome to ensure high humidity conditions and grown for an additional five days at SD growth conditions. To determine CFU counts after leaf infiltration, at 0, 3 and 5 dpi, two leaf discs (5 mm diameter) were taken from each infiltrated leaf with a leaf puncher, placed into 500 μl 10 mM $MgSO_4$ containing two steel beads and homogenized twice for 1 minute at 30 Hz in a TissueLyser II (QIAGEN). Samples were serially diluted and plated on KADO agar plates containing 50 μg/ml kanamycin. After 24h, colonies were counted and log CFU per $cm^2$ leaf material ($\log_{10}$ CFU $cm^{-2}$) was calculated.

### Detection of luminescent bacteria

For spray inoculations, the three most diseased leaves of each plant were sampled 7 dpi and 14 dpi. Each leaf was glued onto an A3 white paper sheet using a grid and covered with a transparent plastic sheet [26]. Next, a light-sensitive X-ray film (Fuji Super RX) was placed on top and left for exposure overnight. The film was developed the next morning and scanned, alongside the leaves, using an A3 flatbed scanner (Epson Expression 12000XL). For 7 dpi, the obtained images were overlayed in Adobe Photoshop 2023/2024 to count the number of luminescent hydathodes per leaf. At 14 dpi, scans of leaves and light-sensitive film were analyzed using the ScAnalyzer image analysis script [58] to determine chlorosis and bacterial colonization. For clipping assays, at 7 dpi, leaves were sampled in the same manner. For analysis, chlorosis and bacterial colonization were analyzed with ScAnalyzer. Bacterial progression through the midvein was measured using Fiji (ImageJ).

### RIL screen and QTL analysis

The Oy-0 × Col-0 RIL population [25] was received from the INRAE/IJPB Versailles stock center. From the 470 RILs in the population, 165 (including the parental Oy-0 and Col-0) were selected for disease assays (S1 Data). For analysis, the mean luminescence index score [26] per leaf (n = 18) for each line was normalized to the score of Oy-0 and used as input for QTL mapping in R using the R/qtl package version 1.66 [27]. LOD scores for each of the 85 markers were determined in a single QTL model using the Expectation-maximization (EM) algorithm. LOD thresholds were determined by

permutation (n = 1000). Explained variance (EV) per QTL was calculated as $EV(\%) = (1 - 10(-2 * LOD/n)) * 100$, where $LOD$ is the LOD score and $n$ is the number of RILs.

## GWA screen and analysis

The HapMap seed collection was received from NASC and propagated (Borevitz lab; NASC ID, N76309; ABRC ID, CS76309). From the 360 lines in the population, 321 accessions (including the controls Oy-0 and Col-0) could be propagated for disease assays (S2 Data). The mean luminescence index score [26] per leaf (n = 18) for each line was normalized to Oy-0. As input for GWA analysis, the data was transformed in a binary score in which 0 = < 50% and 1 = > 50% of Oy-0. These phenotypes were analyzed in the GWA portal [29] using the Imputed Full sequence Dataset, TAIR 9, containing over five million SNPs [67]. P-values were obtained by applying an accelerated mixed model (AMM) [68]. False Discovery Rate (FDR) thresholds were set at $-\log_{10}(p) = 8$ (Bonferroni) and $-\log_{10}(p) = 7.29$ (Benjamini-Hochberg). All SNPs with $-\log_{10}(p)$ values >6 are listed in S2 Data.

## GUS staining and imaging

Transgenic GUS lines were grown for 4–5 weeks at standard SD conditions. Rosettes were taken from soil, washed in double distilled water and submerged in GUS staining solution (0.5 mM $K_3Fe(CN)_6$, 0.5 mM $K_4Fe(CN)_6$, 0.1 M $NaPO_4$ (pH 7.0), 10 mM EDTA, 0.1% (v/v) Triton X-100 and 1 mg/ml X-Gluc in double distilled water) [69] in a 50 ml falcon tube. Rosettes were vacuum infiltrated twice for two minutes and then incubated at 37 °C for two hours to avoid overstaining [69]. To destain, rosettes were incubated in 96% ethanol at room temperature for two days. The destaining solution was replaced twice until all chlorophyll had been removed. Leaves were imaged using a stereomicroscope (Leica MZ FLIII + Nikon Digital Sight DS-Fi2).

## Imaging fluorescent reporter lines

Fluorescent reporter lines were grown for 4–5 weeks using SD conditions (13D/11L). To enable imaging of exposed hydathode tissue, fully expanded leaves were picked and stripped of the epidermal cell layer [70] using household sellotape. Leaves were then imaged with a digital inverted fluorescence microscope (ThermoFisher EVOS fl, RFP filter for tdTomato, YFP filter for mVENUS).

## Protein sequence alignments

Col-0 and Oy-0 whole genome assemblies and annotations [31] were downloaded from NCBI. Coding sequences from both $SUT1^{Col-0}$ and $SUT1^{Oy-0}$ were extracted and translated *in silico*. Predicted protein sequences were then aligned in R with ClustalOmega using the Biostrings (v.2.68.1) and msa (v.1.32.0) packages.

## Quantification and statistical analyses

All data analyses and plotting were done in R (v.4.3.2). All experiments were performed as independent repetitions shown in each figure. Number of biological replicates is mentioned in each figure legend. Data from repetitions was pooled for visualization and statistical analysis.

## Supporting information

**S1 Fig. Manhattan plot of from a screen of 321 accessions present in the HapMap collection.** Highlighted SNPs in orange are located within the coding sequence of *SUT1*. False Discovery Rate (FDR) threshold set at $-\log_{10}(p) = 8$ (Bonferroni). Significantly correlated SNP on Chr 3 is located in the *LAS1* gene (AT3G45130).
(TIF)

**S2 Fig. Alignment of predicted SUT1 protein sequences of the Arabidopsis accessions Col-0 and Oy-0.** Coding sequences of *SUT1[Col-0]* and *SUT1[Oy-0]* were obtained from [31] and translated *in silico* into amino acid sequences. Alignment depicts six predicted amino acid substitutions in SUT1[Oy-0] compared to SUT1[Col-0], of which one is predicted to result in a 3-residue peptide extension at the C-terminus in SUT1[Oy-0].
(TIF)

**S3 Fig. SUT1 resistance in Arabidopsis Col-0 does not depend on TOPP4 function. A)** Number of infected hydathodes per leaf at 7 dpi following spray inoculation with Xcc8004 Δ*xopAC* Tn*7:lux*. Circles depict number of leaves for each score (n = 63 leaves per plant line). The *topp4* knockout shows a slight increase in hydathode infections compared to the Col-0 control. Significance letters from non-parametric Kruskal-Wallis test with Dunn's Post-Hoc test, p-value threshold = 0.05. **B)** Bacterial colonization defined as percentage of total leaf area colonized per leaf following spray inoculation with Xcc8004 Δ*xopAC* Tn*7:lux* (14 dpi, n = 21 leaves per experiment). Bacterial colonization was calculated using ScAnalyzer [58]. The *topp4* knockout does not display increased bacterial spread compared to the wildtype control (Col-0). Significance letters from non-parametric Kruskal-Wallis test with Dunn's Post-Hoc test, p-value threshold = 0.05.
(TIF)

**S4 Fig. *SUT1* is expressed in hydathodes and mesophyll.** Two independent homozygous T3 lines of 4-week-old Arabidopsis *pSUT1::GUS* show GUS enzymatic activity in hydathodes (black arrows) and mesophyll tissue throughout the leaf. Leaves were incubated in GUS staining solution for two hours.
(TIF)

**S5 Fig. *SUT1* is expressed in epithem cells within hydathodes.** Three independent T2 lines of 4-week-old Arabidopsis *pSUT1::NLS-3×mVENUS* show nuclear accumulation of mVENUS in the mesophyll and epithem. Epithem cells can be distinguished from mesophyll cells in the bright-field image as a cell dense area present on the leaf margin connected to the vasculature (white dashed line).
(TIF)

**S6 Fig. *RKS1/ZAR1* immunity against Xcc in the leaf vasculature. A)** Disease symptoms and bacterial luminescence in resistant Col-0 and susceptible *zar1–1* knockout lines, 7 dpi following clip inoculation with Xcc8004 WT Tn*5:lux*. **B,C)** Chlorotic leaf area (**B**) and bacterial spread along the midvein (**C**) 7 dpi following clip inoculation with Xcc8004 WT Tn*5:lux* on Arabidopsis Col-0, *bak1–5;bkk1–1*, *zar1–1*, *rks1–1* and Oy-0 (n = 12 leaves). ZAR1-mediated resistance against Xcc is effective in the vasculature of Col-0 and *bak1–5;bkk1–1* when the hydathodes are bypassed and the bacteria are introduced using leaf clipping. Significance letters from a two-way ANOVA with Tukey Post-Hoc test, p-value threshold = 0.05.
(TIF)

**S7 Fig. Confirmation of *rks1/sut1* double knockout mutants in Col-0 background. A,B)** Schematic overview of two guide RNAs targeting the first exons of each target gene, *RKS1* (**A**) and *SUT1* (**B**), in Col-0. **C)** Schematic overview of resulting homozygous (when single sequence is shown) or biallelic (when two sequences are shown) indels (green N's) introduced at each gRNA target sode (g1 and g2) in two individual T1 mutant lines (#1 and #2). Both lines, *rks1/sut1* #1 and #2, show confirmed early STOP codons in both alleles of both target genes, as indicated on the right.
(TIF)

**S1 Table. List of oligonucleotides, plasmids, plant lines and bacterial strains used in this study.**
(XLSX)

**S1 Data. Raw data of RIL population genetic structure and disease scoring.** Disease scores of each line used in the RIL screen. Column A depicting average disease index score relative to the Oy-0 control. Column B depicts

line numbers used. Columns D-CJ show parental genotype (A = Col-0, B = Oy-0, C = *NA*) for each marker in each line.
(XLSX)

**S2 Data. Raw data on disease scoring of HapMap population for GWAS.** File contains disease scores for each line in the HapMap screen. Column B shows the line name. Column D shows ABRC ID numbers of each line. Column F depicts average disease index score relative to the Oy-0 control. The second tab shows all SNPs that have a $-\log_{10}$ (p) score of >6.
(XLSX)

## Acknowledgments

We would like to thank Ludek Tikovsky and Harold Lemereis for the excellent care of our plants in the greenhouse. We also want to thank Manon Richard for previous work on fluorescent reporter lines in Arabidopsis. Lastly, we want to thank Niko Geldner, Laurent Noël and Gail Preston for sharing materials.

## Author contributions

**Conceptualization:** Nanne W. Taks, Marieke van Hulten, Jeroen A. van Splunter-Berg, Misha Paauw, Sebastian Pfeilmeier, Harrold A. van den Burg.

**Data curation:** Nanne W. Taks, Marieke van Hulten.

**Formal analysis:** Nanne W. Taks.

**Funding acquisition:** Harrold A. van den Burg.

**Investigation:** Nanne W. Taks, Marieke van Hulten, Jeroen A. van Splunter-Berg, Sayantani Chatterjee, Floris D. Stevens, Misha Paauw.

**Methodology:** Nanne W. Taks, Marieke van Hulten, Jeroen A. van Splunter-Berg.

**Project administration:** Nanne W. Taks, Harrold A. van den Burg.

**Resources:** Jeroen A. van Splunter-Berg, Floris D. Stevens, Harrold A. van den Burg.

**Supervision:** Sebastian Pfeilmeier, Harrold A. van den Burg.

**Validation:** Nanne W. Taks, Marieke van Hulten.

**Visualization:** Nanne W. Taks.

**Writing – original draft:** Nanne W. Taks, Sebastian Pfeilmeier, Harrold A. van den Burg.

**Writing – review & editing:** Nanne W. Taks, Sebastian Pfeilmeier, Harrold A. van den Burg.

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
