## [Decision Letter · Decision Letter 0]

PPATHOGENS-D-25-00503

*Arabidopsis* CNL receptor SUT1 confers immunity in hydathodes against the vascular pathogen *Xanthomonas campestris* pv. *campestris*

PLOS Pathogens

Dear Dr. van den Burg,

Thank you for submitting your manuscript to PLOS Pathogens. After careful consideration, we feel that it has merit but does not fully meet PLOS Pathogens's publication criteria as it currently stands. Therefore, we invite you to submit a revised version of the manuscript that addresses the points raised during the review process.

My apologies for again for the delay in this response, which was again due to reviewer-based issues out of my control. As you can see, reviewer 1 was satisfied with the revisions. I request that you submit a revised version that addresses the valuable points raised by reviewer 2 (a new reviewer).

Please submit your revised manuscript within 30 days Jun 24 2025 11:59PM. If you will need more time than this to complete your revisions, please reply to this message or contact the journal office at plospathogens@plos.org. Please include the following items when submitting your revised manuscript:

We look forward to receiving your revised manuscript.

Kind regards,

David Mackey

Academic Editor

PLOS Pathogens

Savithramma Dinesh-Kumar

Section Editor

PLOS Pathogens

Sumita Bhaduri-McIntosh

Editor-in-Chief

PLOS Pathogens

orcid.org/0000-0003-2946-9497

Michael Malim

Editor-in-Chief

PLOS Pathogens

orcid.org/0000-0002-7699-2064

**Journal Requirements:**

4) We notice that your supplementary Figures are included in the manuscript file. Please remove them and upload them with the file type 'Supporting Information'. Please ensure that each Supporting Information file has a legend listed in the manuscript after the references list.

Potential Copyright Issues:

- Please confirm (a) that you are the photographer of Figures 1A, 2C, 3A, 4A, 5A, 5D, S4, and S6, or (b) provide written permission from the photographer to publish the photo(s) under our CC BY 4.0 license.

- Figure 5. Please confirm whether you drew the images / clip-art within the figure panels by hand. If you did not draw the images, please provide (a) a link to the source of the images or icons and their license / terms of use; or (b) written permission from the copyright holder to publish the images or icons under our CC BY 4.0 license. Alternatively, you may replace the images with open source alternatives. See these open source resources you may use to replace images / clip-art:

6) Please ensure that the funders and grant numbers match between the Financial Disclosure field and the Funding Information tab in your submission form. Note that the funders must be provided in the same order in both places as well. State what role the funders took in the study. If the funders had no role in your study, please state: "The funders had no role in study design, data collection and analysis, decision to publish, or preparation of the manuscript.".

**Reviewers' Comments:**

Reviewer's Responses to Questions

**Part I - Summary**

Reviewer #1: Authors have satisfactorily addressed my concerns. The revised manuscript is considerably improved.

Reviewer #2: This study identifies a locus encoding a novel coiled-coil nucleotide-binding leucine-rich repeat (CNL) immune receptor gene in Arabidopsis thaliana Col-0, named SUPPRESSOR OF TOPP4 1 (SUT1). The authors develop a creative, screening approach that overlays typical resistance phenotyping with observing bacterial colonization with luciferase and determine that the SUT1 locus is frequently associated with resistance against the vascular pathogen Xanthomonas campestris pv. campestris (Xcc) specifically at the hydathodes, the bacteria's entry point. This would not be easily observed with traditional phenotyping and provides an important example where researchers could use this approach across many plant-microbe systems. Through genetic mapping and association studies, mutations were enriched, although not statistically significant at SUT1. Genetic analysis of publicly available and edited SUT1 mutant lines further supports the key role of this gene in early-stage immunity beyond the previously characterized RKS1/ZAR1 pathway. They could not complement SUT1. While SUT1 is expressed in hydathode epithem cells and throughout the leaf, its resistance function is primarily effective at the hydathodes, contrasting with RKS1/ZAR1, which provides broader resistance in xylem and mesophyll tissues, highlighting a tissue-specific deployment of distinct immune mechanisms against Xcc infection in Arabidopsis.

**Part II – Major Issues: Key Experiments Required for Acceptance**

Reviewer #1: (No Response)

Reviewer #2: 1) They could not complement SUT1, which raises questions about the definitive causal link or potential complexities of the gene's function. They take on the valiant, honest effort to include additional mutations, but there are serious concerns that this could be occurring in the non-coding space at the same location. Moreover it could suggest alternative splicing or some other regulation (e.g. postranscriptional) that permit function in the hydathodes and not along the veins for vascular infection. There was a missed opportunity to potentially discuss how SUT1 may not function in the xylem because the tissue is comprised of dead cells, but in Figure 3A, the GUS staining is all along the veins. Overall there current work just denotes that the SUT1 locus is important for hydathode immunity.

2) The reporter gene expression studies in Figure 3 provide evidence for SUT1 promoter activity in hydathodes, which aligns with its proposed role in early resistance. In Figure 3, the authors however appear to have omitted several crucial controls for imaging or defining this phenomenon. They lack a quantifiable output (i.e. the number of total hydathodes express this compared to a control). They largley emphasize the importance of SUT1 based on the localization. It is interesting that the SUT1 promoter is active at the hydathode, and it appears to not be preferential. Looping back to a previous point above: why would the authors hypothesize that SUT1 is not effective in the vascular system even though it is visibly expressed along the veins in panels A? There could be valuable discussion around the xylem being comprised of dead cells, but curiously it is still expressed along the veins, which likely is localized to the xylem parenchyma. The authors surprising do not discuss this. Moreover the authors use an extremely high cell density for leaf clipping in the virulence assay in Figure 4, which at first glance may suggest it is inneffective in the vascular system, but their OD600 = 0.1, which by guess is equivalent to 10^8 CFU/mL. Overall, could their results be masked by the high cell density from leaf clipping? When they perform direct infiltration, the team uses 0.0001 OD600, which is dramatically lower and also a form of direct inoculation like leaf clipping. They may see quantifiable differences with a lower cell density. The pPUP1 comparison is also an interesting control but still doesn’t resolve the issues viewed from their GUS and mVENUS expression, which are expressed everywhere. Why did they choose pPUP1 based on the publication cited (ref 16)? They could use this as a control to quantify their GUS results. Would the expression of pSUT1::GUS look similar to p35S::GUS? This seems like an important control. Overall it would be useful to quantify the output here. Maybe a slightly better experiment would be to co-express pPUP1::RCl2a-tdTomato and pSUT1::NLS-3xmVENUS in the same line, but I hesitate to suggest this as a must as it would delay publication further. Such controls would be expected in a protein gel/blot comparison of expression. If they cannot make the line, an alternative approach could involve measuring pPUP expression relative to pSUT1 expression from excised hydathodes if pPUP serves as an epithem control compared to other tissues with qPCR.

**Part III – Minor Issues: Editorial and Data Presentation Modifications**

Reviewer #1: (No Response)

Reviewer #2: Line 65: It is an obvious introduction point to bring in innate immunity and flagellin, but the flagellin literature is quite unclear around this being a major innate immune signal. Multiple reports suggests high variation in flagellin perception. This would highlight an important gap or inconsistency compared to Pseudomonas literature and also emphasize to a non-expert reader the variation in the literature. (e.g. doi: 10.1094/MPMI-08-21-0211-R, doi: 10.7717/peerj.18204)

Line 266 the “r” is not italicized

overall: "P" is capitalized for promoter. "p" is for plasmid.

overall: Do they have normalization data to show that lux output in Fig 2F for bacterial colonization and Fig 4C for bacterial movement actually have a relationship with CFU for these particular assays?

PLOS authors have the option to publish the peer review history of their article (what does this mean? ). If published, this will include your full peer review and any attached files.

**Do you want your identity to be public for this peer review?** For information about this choice, including consent withdrawal, please see our Privacy Policy .

Reviewer #1: No

Reviewer #2: No

**Figure resubmission:**
---

## [Editor Report · Decision Letter 1]

Dear Dr van den Burg,

We are pleased to inform you that your manuscript '*Arabidopsis* CNL receptor SUT1 confers immunity in hydathodes against the vascular pathogen *Xanthomonas campestris* pv. *campestris* ' has been provisionally accepted for publication in PLOS Pathogens.

Best regards,

David Mackey

Academic Editor

PLOS Pathogens

Savithramma Dinesh-Kumar

Section Editor

PLOS Pathogens

Sumita Bhaduri-McIntosh

Editor-in-Chief

PLOS Pathogens

orcid.org/0000-0003-2946-9497

Michael Malim

Editor-in-Chief

PLOS Pathogens

orcid.org/0000-0002-7699-2064
---

## [Editor Report · Acceptance letter]

Dear Dr van den Burg,

We are delighted to inform you that your manuscript, "Arabidopsis CNL receptor SUT1 confers immunity in hydathodes against the vascular pathogen *Xanthomonas campestris* pv. *campestris* ," has been formally accepted for publication in PLOS Pathogens.

Best regards,

Sumita Bhaduri-McIntosh

Editor-in-Chief

PLOS Pathogens

orcid.org/0000-0003-2946-9497

Michael Malim

Editor-in-Chief

PLOS Pathogens

orcid.org/0000-0002-7699-2064